# Exploring the immune microenvironment in small bowel adenocarcinoma using digital image analysis

Fatima Abdullahi Sidi[1‡], Victoria Bingham[1‡], Stephen McQuaid[1,2,3], Stephanie G. Craig[1], Richard C. Turkington[4], Jacqueline A. James[1,2,3], Matthew P. Humphries[1,5,6‡*], Manuel Salto-Tellez[1,2,7‡*]

**1** Precision Medicine Centre of Excellence, The Patrick G. Johnston Centre for Cancer Research, Queen's University Belfast, Antrim, United Kingdom, **2** Cellular Pathology, Belfast Health and Social Care Trust, Belfast City Hospital, Lisburn Road, Belfast, Antrim, United Kingdom, **3** Northern Ireland Biobank, The Patrick G. Johnston Centre for Cancer Research, Queen's University Belfast, Antrim, United Kingdom, **4** The Patrick G. Johnston Centre for Cancer Research, Queen's University Belfast, Antrim, United Kingdom, **5** Leeds Teaching Hospitals NHS Trust, Leeds, West Yorkshire, United Kingdom, **6** University of Leeds, St James' University Hospital, Leeds, West Yorkshire, United Kingdom, **7** Division of Molecular Pathology, The Institute for Cancer Research, London, Greater London, United Kingdom

‡ FAS and VB are joint first authors. MPH and MST are joint senior authors.
* matthew.humphries2@nhs.net (MPH); m.salto-tellez@qub.ac.uk (MST)

**Data Availability Statement:** The data underlying the results presented in the study are available from The Precision Medicine Centre of Excellence (PMC) at Queen's University Belfast: Queen's

## Abstract

### Background

Small bowel adenocarcinoma (SBA) is a rare malignancy of the small intestine associated with late stage diagnosis and poor survival outcome. High expression of immune cells and immune checkpoint biomarkers especially programmed cell death ligand-1 (PD-L1) have been shown to significantly impact disease progression. We have analysed the expression of a subset of immune cell and immune checkpoint biomarkers in a cohort of SBA patients and assessed their impact on progression-free survival (PFS) and overall survival (OS).

### Methods

25 patient samples in the form of formalin fixed, paraffin embedded (FFPE) tissue were obtained in tissue microarray (TMAs) format. Automated immunohistochemistry (IHC) staining was performed using validated antibodies for CD3, CD4, CD8, CD68, PD-L1, ICOS, IDO1 and LAG3. Slides were scanned digitally and assessed in QuPath, an open source image analysis software, for biomarker density and percentage positivity. Survival analyses were carried out using the Kaplan Meier method.

### Results

Varying expressions of biomarkers were recorded. High expressions of CD3, CD4 and IDO1 were significant for PFS ($p = 0.043$, $0.020$ and $0.018$ respectively). High expression of ICOS was significant for both PFS ($p = 0.040$) and OS ($p = 0.041$), while high PD-L1 expression in tumour cells was significant for OS ($p = 0.033$). High correlation was observed between PD-L1 and IDO1 expressions (Pearson correlation co-efficient = 1) and

University Belfast. Health Sciences Building. 97 Lisburn Road. Belfast. BT9 7AE. Tel: +44(0)28 9097 2293. m.salto-tellez@qub.ac.uk.

**Funding:** This study was funded by a Cancer Research UK (CRUK) Accelerator Grant (A20256) to JJ and MST. CRUK (https://www.cancerresearchuk.org/) had no role in study design, data collection and analysis, decision to publish, or preparation of the manuscript. The Northern Ireland Biobank has received funds from HSC Research and Development Division of the Public Health Agency in Northern Ireland.

**Competing interests:** I have read the journal's policy and the authors of this manuscript have the following competing interests: Manuel Salto-Tellez is a scientific advisor to Mindpeak and Sonrai Analytics, and has received honoraria recently from BMS, MSD and Incyte. None of these disclosures are related to this work. The remaining authors declare no potential conflicts of interest.

**Abbreviations:** FFPE, Formalin Fixed Paraffin Embedded; ICI, Immune Checkpoint Inhibitor; ICOS, Inducible T-cell COStimulator; IDO1, Indoleamine 2, 3-DiOxygenase 1; OS, Overall Survival; PD-L1, Programmed cell Death Ligand 1; PFS, Progression-Free Survival; SBA, Small Bowel Adenocarcinoma; TMA, Tissue Micro-Array.

subsequently high IDO1 expression in tumour cells was found to be significant for PFS ($p = 0.006$) and OS ($p = 0.034$).

## Conclusions

High levels of immune cells and immune checkpoint proteins have a significant impact on patient survival in SBA. These data could provide an insight into the immunotherapeutic management of patients with SBA.

## Introduction

### Aetiology and incidence

Small bowel cancers (SBCs) are malignant lesions found in the small intestine, which makes up about 80% of the intestinal tract's length. Although an uncommon neoplasm, SBCs result in approximately 1700 new cancer cases every year in the UK, accounting for less than 1% of total cancer cases. In the last two decades, the incidence rates for SBCs in the UK have risen by about 163%. Though occurring more commonly in men, women have seen larger increased incidence over that time period [1–6]. Potential influences on the rising incidence of SBC include increased use of imaging tests, inflammatory bowel diseases, changes in the microbiome, environmental factors, and genetic predisposition [7–11].

Small bowel adenocarcinoma (SBA) is the most common histological type of small bowel tumour, being more common in the duodenum, and is associated with late stage diagnosis, poor prognosis and high mortality rates [1, 2, 12, 13]. Five year overall survival for early stage disease has been reported to be between 35%– 80%, with 8%– 47% for locally advanced disease and 5% for disseminated disease [14]. The management of SBA clinically is similar to that of colorectal cancer (CRC) adenocarcinomas. Differences have been observed in incidence and disease progression within the two cancer groups, with SBA having increasing incidence and poorer outcome compared to CRC [14–16]. The difference may be due to anatomical, genetic, clinical, and treatment factors [15, 17].

### The tumour microenvironment in gastrointestinal tumour biology

Studies carried out on the tumour and immune microenvironment have shown infiltration of tumour regions by T-lymphocytes have been shown to improve survival in oesophageal, gastric and colorectal cancers among others [14, 18, 19]. Immune checkpoint inhibition by the interaction of Programmed Cell Death Protein-1 (PD-1) and Programmed Cell Death Ligand 1 (PD-L1) in the tumour microenvironment has been shown to impact disease progression in many cancers including oesophageal cancer as well as microsatellite instability-high (MSI-H) gastric and colorectal cancers [13, 14, 19, 20].

In small bowel neuroendocrine tumours (SB NETs), studies have shown immune infiltration of tumour regions and expression of PD-L1 on tumour cells. However, both T-cell infiltration and PD-L1 positivity were found to have no significance on patient survival [21, 22]. Presence of tumour infiltrating lymphocytes (TILs) and PD-L1 positivity have been reported in SBA with favourable prognostic impact seen in terms of survival. This has led to the proposal for use of such data in the development of immune checkpoint therapy strategies for SBA [13, 14, 20].

Several methodologies have been used to study the microenvironment of SBA, including flow cytometry, and gene expression profiling [14, 15]. Comprehensive studies have used

immunohistochemistry (IHC) to explore the microenvironment of SBA. For example, a study by Wang et al. analysed the expression of immune-related markers, including PD-L1, CD3, CD4, CD8, FoxP3, and CD163 [23]. Another study by Liu et al. used IHC to analyse the expression of tumour-infiltrating lymphocytes and immune checkpoint markers, including PD-L1 and CTLA-4 [24]. While microsatellite instability (MSI), mismatch repair deficiency (MMR-d) and tumour mutation load (TML) status have been shown to have favourable prognosis in terms of cancer-specific survival [13, 14, 25, 26]. In a previous study, we highlighted the molecular profile of SBA and identified genes and pathways that contributed significantly to the pathogenesis of the disease as well as survival outcome [2].

## Study aim

In this study, we investigate the immune composition of the tumour microenvironment in SBA using a subset of immunomodulatory biomarkers which include the T-cell markers (CD3, CD4, and CD8), macrophage marker (CD68), and immune checkpoint biomarkers (IDO1, ICOS, LAG3 and PD-L1) as well as their prognostic impact in terms of disease outcome and patient survival.

## Materials and methods

### Ethical approval, patient sample and clinicopathological data

Ethical approval for the study was obtained from the Northern Ireland Biobank (Ethics 11/NI/0013, NIB13-0067 & NIB15-0168) [27]. Tissue samples were obtained from 25 patients with SBA diagnosed between 2002 and 2013 and relevant clinicopathological information was received from the Belfast Health and Social Care Trust. De-anonymised samples were made available for analysis in the form of formalin fixed, paraffin embedded (FFPE) tissue blocks. The results are reported in adherence to the REMARK criteria [28].

### Tissue micro-array construction and immunohistochemistry

Tissue Micro-Arrays (TMAs) were constructed by taking three representative tumour areas from each donor FFPE tissue block on a Beecher MTA1 (Beecher Instruments Inc., WI) as previously described [2]. Briefly, representative tumour area selection involves a rigorous selection process by an expert pathologist to identify a tumour area which reflects the characteristics of the tumour as a whole. This process typically requires examination of multiple tissue sections from the donor tissue blocks. Exact areas are annotated and tissue cores obtained in triplicate ensuring that the tumour characteristics are consistent. To control for tumour heterogeneity, section review is conducted consistently to ensure the representativeness of the TMA is retained as it is cut for each study. The manual arrayer was used to extract 1mm cores for insertion into recipient blocks. For IHC 3 micron sections were taken from each TMA slide and stained using previously validated antibodies for CD3, CD4, CD8, CD68, ICOS, IDO1, LAG3 and PD-L1. Staining conditions for all antibodies are listed in Table 1. Automated IHC was performed using a Leica BOND RX (Leica Biosystems, USA) ensuring staining consistency across all samples.

### Slide scanning and digital image analysis

The TMA slides were scanned on an Aperio AT2 digital scanner (Leica Biosystems, Milton Keynes, UK) and stored on a secure network server. Image analysis was carried out using QuPath open-source software (version 0.2.2). Images were imported into QuPath, consisting of triplicate TMAs per biomarker. TMAs were de-arrayed prior to analysis and individual

**Table 1. Antibody staining conditions.**

| Marker | Clone | Company/Catalogue Number | Epitope Retrieval | Dilution + Incubation | Detection |
|--------|-------|--------------------------|-------------------|-----------------------|-----------|
| CD3 | 2GV6 | Roche/790-4341 | CC1 32mins | Neat for 16min @ 37C | Optiview DAB |
| CD4 | SP35 | Roche/790-4423 | CC1 32mins | Neat for 16min @ 37C | Ultraview DAB |
| CD8 | C8/144B | Dako/M7103 | ER2 20min | 1/200 for 15min @RT | Bond Polymer Refine + Enhancer |
| CD68 | KP1 | Roche/790-2931 | CC1 60mins | Neat for 16min @ 37C | Optiview DAB |
| ICOS | D1K2T | Cell Signaling/89601 | ER2 20min | 1/400 for 15min @RT | Bond Polymer Refine +Enhancer |
| IDO1 | D5J4E | Cell Signaling/86630 | ER2 20min | 1/400 for 15min @RT | Bond Polymer Refine +Enhancer |
| LAG3 | 17B4 | LSBio/LS-B2237 | ER1 20min | 1/100 for 15min @ RT | Bond Polymer Refine +Enhancer |
| PD-L1 | SP263 | Roche/790-4905 | CC1 64mins | Neat for 16min @ 37C | Optiview DAB |

cores were checked for suitability, ensuring areas of tissue folds, necrosis and artefacts were manually removed in a supervised quality control process as previously described [29]. Areas of tissue were detected within each core and for the biomarkers CD3, CD4, CD8, CD68, ICOS, IDO1 and LAG3, positive cell detection was carried out using pathologist-guided DAB thresholds based on the staining intensity and pattern of staining (either cell or nucleus) to identify the total number of positive cells as well as the densities per mm$^2$. Thresholds were standardised and consistently applied across all images, for each marker, following pathologist confirmation of sensitivity and specificity over several images. For PD-L1, percentage positivity within the tumour epithelial cell compartment as well as the whole core was assessed following application of a tumour-stroma classifier, based in QuPath. Variables and parameters used to build a tumour stroma classifier include; image annotation of tumour stroma areas (MPH), reviewed and confirmed by expert clinicians (SMcQ and MST); intensity and texture feature extraction of annotated areas, including pixel values and Haralick features in the annotated areas. Following cell detection, features such as cell area, perimeter and circularity features were used to train the random forest machine learning algorithm to classify tumour and stroma cells into distinct compartments. Lastly, model evaluation was subjectively evaluated for accuracy, sensitivity, and specificity over multiple images (SMcQ and MST). A PD-L1 positive cell was identified by either a tumour or stroma cell showing complete or partial membrane staining. Evaluation of the expression of each biomarker as well as setting of positivity thresholds were confirmed by an experienced pathologist, as reported previously [30–32].

## Data and statistical analysis

The measurements obtained from QuPath were exported into a spreadsheet (Microsoft Excel 2016) where the average scores for each patient were calculated based on the triplicate TMA cores used per biomarker for each patient. The density of positive cells per mm$^2$ for each biomarker was obtained and dichotomised using the median number in each category as a cut-off. Survival analysis was conducted using the Kaplan-Meier method and log-rank test for progression-free and overall survival (PFS and OS). A $p$-value of $<0.05$ was considered statistically significant. Statistical analyses were carried out using IBM$^®$ SPSS Statistics (version 26) and R using RStudio (version 1.3.1093). Our data set underlying the results described can be found in S1 File.

## Results

### Clinicopathological characteristics

Samples from 25 patients were available for analysis (Table 2). Of these, 15 were female and 10 were male with ages ranging from 32 to 85 years. The jejunum was found to be the most

**Table 2. Summary of clinicopathological data.**

| Characteristic | N (Total = 25 (100%)) |
|---|---|
| Sex: | |
| Male | 10 (40%) |
| Female | 15 (60%) |
| Age: | |
| Minimum | 32.00 |
| Median | 61.00 |
| Mean | 61.04 |
| Maximum | 85.00 |
| Tumour Site: | |
| DJ Flexure | 1 (4%) |
| Duodenum | 6 (24%) |
| Ileum | 6 (24%) |
| Jejunum | 8 (32%) |
| Small Intestine NOS | 4 (16%) |
| Extent at Presentation: | |
| Localized/Resected | 18 (72%) |
| Metastatic | 5 (20%) |
| Unknown | 2 (8%) |
| T Stage: | |
| T3 | 10 (40%) |
| T4 | 1 (4%) |
| T4b | 14 (56%) |
| N Stage: | |
| N0 | 5 (20%) |
| N1 | 15 (60%) |
| N2 | 2 (8%) |
| Nx | 3 (12%) |
| M Stage: | |
| M1 | 5 (20%) |
| Mx | 19 (76%) |
| Unknown | 1 (4%) |
| Adjuvant Therapy: | |
| Yes | 10 (40%) |
| No | 8 (32%) |
| Unknown | 7 (28%) |

common tumour site in 8 patients, followed by the duodenum and ileum (six patients each), one patient had tumours located at the duodeno-jejunal junction and for four patients the location was not specified. Survival data were available for all patients with a median OS of 39 months.

## Biomarker expression

The T-cell markers CD3, CD4, CD8 were specific to subsets of lymphocytes predominantly located in stromal tissue adjacent to tumour epithelium, but with varying amounts of intra-tumoural expressing T-cells. CD68 was expressed on the varying populations of macrophages

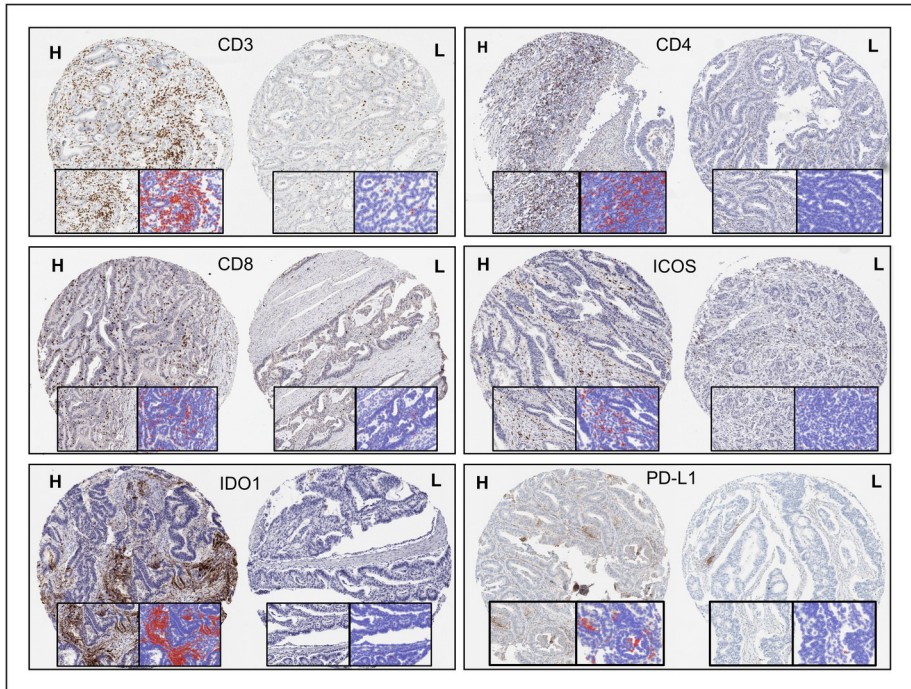

**Fig 1. Biomarker expression across whole TMA cores.** High (H) and low (L) levels of positivity depicted for each biomarker as well as the QuPath cell detection mask positive cells (red) and negative cells (blue).

present in the tissue cores. The immune checkpoint biomarkers ICOS and LAG3 were exclusively expressed on sub-populations of lymphocytes in the stroma. Expression of IDO1 was more variable, with expression on sub-populations of lymphocytes in the stroma but also occasional strong expression on intra-tumoural nests of epithelial cells. When present in a case, PD-L1 was expressed on tumour epithelial cell membranes and on stromal lymphocytes and macrophages. Examples of optimised and validated staining for various biomarkers on TMA cores are shown in Fig 1.

## Biomarker scoring

Whole core expression was assessed for each single IHC biomarker using a robust digital pathology workflow as previously described [19]. The densities in $mm^2$ of positive cells for CD3 ranged from 42.6 to 8423.0 (median = 409.4), 0.8 to 3411.7 (median = 34.0) for CD4, 14.5 to 6450.3 (median = 235.5) for CD8, 23.7 to 3495.0 (median = 460.1) for CD68, 1.0 to 3569.9 (median = 39.2) for ICOS, 0.0 to 6604.6 (median = 44.9) for IDO1 and 0.0 to 826.8 (median = 63.8) for LAG3. For PD-L1, percentage expression was calculated based on positive cells within the whole core for all cells detected as well as within the tumour epithelium for all tumour cells detected. The scores ranged from 0.0% to 53.1% in the whole core and 0.0% to 12.7% in the tumour epithelium, and were dichotomised based on <1% or ≥1% PD-L1 expression.

## Survival analysis

Kaplan Meier survival analysis showed that high CD3, CD4 and IDO1 were significant for PFS ($p$ = 0.043, 0.020 and 0.018 respectively) but not OS. High expression of ICOS showed

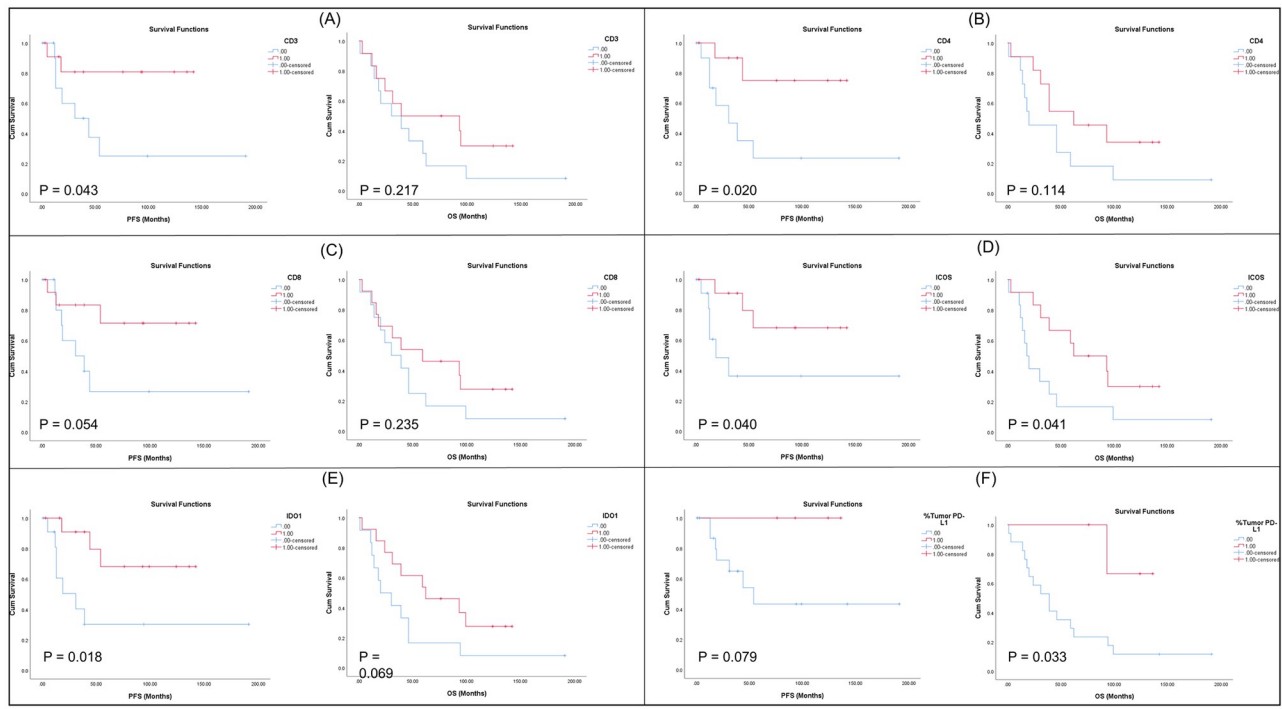

**Fig 2. Kaplan Meier survival plots.** Associations between high biomarker expression and survival in terms of PFS and OS. A, B, & E) High expression of CD3, CD4 & IDO1 respectively associated with better PFS. D) High expression of ICOS associated with better PFS & OS. F) Better OS associated with high tumour PD-L1 expression. C) Trend towards better PFS for high CD8 expression.

significance in terms of both PFS ($p = 0.040$) and OS ($p = 0.041$) while high PD-L1 expression in tumour was associated with better OS ($p = 0.033$). High expression of CD8 showed a trend towards significance for PFS but failed to achieve significance for OS while high CD68 and LAG3 expression were not significant for PFS or OS (Fig 2).

The correlation between biomarker expressions was assessed for all eight biomarkers in a correlation matrix using Pearson correlation test. The highest correlation was observed between PD-L1 and IDO1 with a correlation co-efficient of 1, while the lowest correlation was observed between LAG3 and all the other biomarkers (Fig 3).

For IDO1, a further assessment of expression on tumour epithelium was the most optimal method using a ($0 = 0\%$, $1 = 1$–$20\%$, $2 \geq 20\%$) scoring system. Kaplan Meier survival analysis showed that high expression of IDO1 on tumour cells (score of 2) was associated with better survival in terms of PFS ($p = 0.006$) and OS ($p = 0.034$) (Fig 4).

## Discussion

From this study, we are able to obtain a picture of the immune microenvironment in SBA. With the help of digital image analysis, we assessed the expression of immune and macro-phages markers (CD3, CD4, CD8, and CD68) at a whole core level. Furthermore we assessed the presence of immune checkpoint biomarkers (IDO1, ICOS, LAG3 and PD-L1) on tumour epithelial and inflammatory cells across the sample tissues. The various biomarkers, where present, were scored in terms of density per mm$^2$ or percentage expression for PD-L1.

In this patient cohort, we show that the presence of immune biomarkers in the tumour microenvironment, especially CD3 and CD4, had significant impact on PFS. High expression

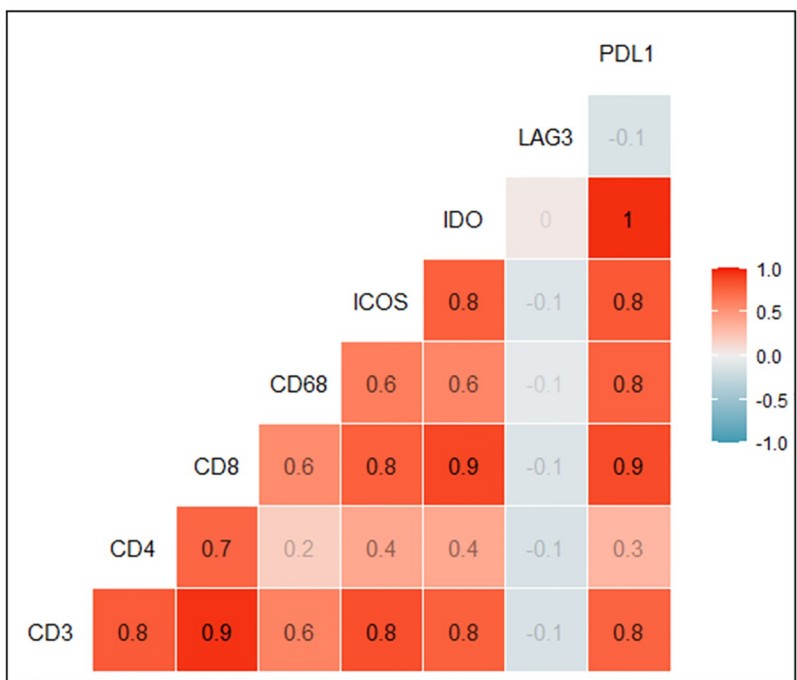

**Fig 3. Correlation plot displaying the relationship between biomarkers using the Pearson correlation test.** Highest correlation (red) is seen between PD-L1 and IDO1 and the lowest correlations (blue) are seen between LAG3 and all other biomarkers. The absolute number of positive cells averaged for each marker was used in the analysis, with the exception of PD-L1 which was categorised in line with clinical scoring criteria.

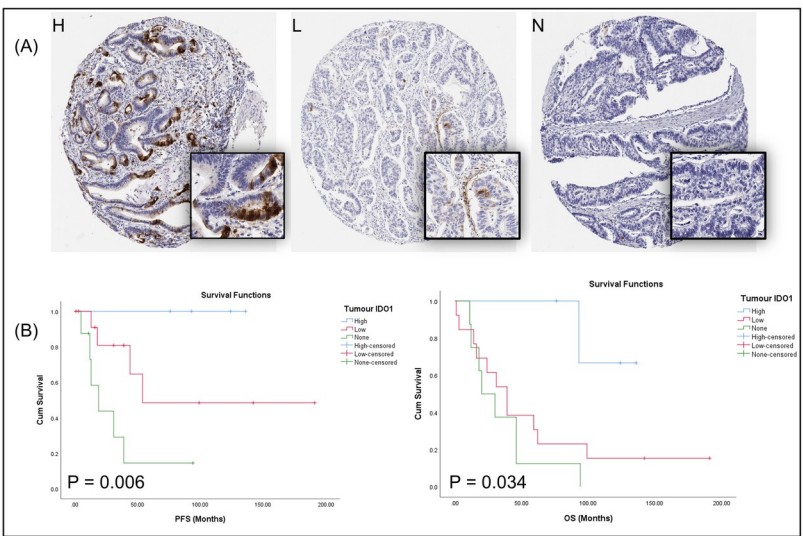

**Fig 4. IDO-1 assessment.** A) Expression of IDO1 on tumour epithelial cells based on staining intensity. High expression (H) is seen depicted as ≥ 20% and scored as 2. Low expression (L) and no expression (N) categorized as 1 (1–20%) and 0 (0%) respectively. B) Kaplan Meier survival plots demonstrate that high expression of IDO1 on tumour epithelium is significantly associated with improved PFS & OS.

of the immune checkpoint inhibitors (ICIs) IDO1, ICOS and PD-L1 also showed favourable outcome for survival. A test of correlation carried out for all biomarkers showed the highest correlation between biomarkers was with PD-L1 and IDO1. As high PD-L1 expression on tumour cells was associated with better OS, we carried out a similar analysis for IDO1 and found that high expression of IDO1 on tumour epithelial cells was also associated with better PFS and OS.

Similar findings to the present study in terms of immune cell infiltration into the tumour microenvironment in SBCs have been reported, including the variation of expression between high and low lymphocyte densities across samples. Donisi et al. for example, also showed CD8 T-cell data is suggestive that T-cell infiltration may play a protective role [33]. The same studies also reported PD-L1 expression on tumour epithelial cells as well as immune cells although using different PD-L1 scoring methods and cut-offs [13, 14, 20, 21]. Our PD-L1 scoring method and cut-off was similar to that used by Thota et al., who also found high levels of PD-L1 expression to be associated with high levels of CD3, CD4 and CD8 [20]. Wirta et al. used a cut-off of 5% to determine PD-L1 positivity and an immune cell score (ICS) for combined CD3 and CD8 positivity, and found a strong relationship between high ICS and PD-L1 expression in TILs [14].

Interestingly, despite reporting high expressions of T cells across samples, only CD3 and CD4 were found to have prognostic significance and only CD3 has been shown to be a favourable prognostic factor alone [13], or in association with CD8 in a combined ICS [14]. In the present study CD8 expression alone did not show favourable statistical significance. However. PD-L1 positivity in tumour epithelial cells alone was found to be significant for survival but not when expression on all cell phenotypes was assessed. Other studies found that PD-L1 expression in tumour cells was not significant in terms of prognosis, however, Wirta et al. found PD-L1 expression in immune cells to be an independent prognostic biomarker for survival in the presence of MSI and MMR-d status [13, 14].

In their study on GI cancers in a cohort of 4,125 patients (of which 143 were SBA cases), Salem et al. found varying expressions of PD-L1 on tumour cells in the different cancer subtypes. Oesophageal squamous and anal cancers had the highest PD-L1 expression while colorectal cancers had the lowest PD-L1 expression. 27 (19%) of the SBA cases had positive PD-L1 expression at 1% cut-off while only 15 (10%) had PD-L1 positivity at 5% cut-off [25]. High levels of tumour immune cell infiltrates have been reported in CRC and were associated with improved patient outcome, with PD-L1 expression reported to impact survival in the presence of MSI status [34–38]. For oropharyngeal, oesophageal and gastric cancers, increased T cell infiltrates have been associated with favourable prognosis, however, PD-L1 expression on immune cells was seen to impact survival rather than tumour cell expression [23, 39–45].

To our knowledge, PD-L1 is the only ICI protein to be studied so far in relation to survival in SBAs. In this study, we found that ICOS and IDO1 had a significant impact on survival outcomes. Interestingly, increased expression of IDO1 has been reported in CRC and is associated with worse prognosis and poor disease outcome, with its inhibition shown to improve patient survival [46, 47]. IDO1 expression was seen in oesophageal adenocarcinoma but had no impact on survival [19]. Blakely et al. found a high expression of both PD-L1 and IDO1 in gastrointestinal stromal tumours (GIST). They also found that in the presence of high TIL counts (CD3 and CD8), GISTs with high PD-L1/IDO1 positivity showed significant smaller tumour size [48]. ICOS positivity on TILs in gastric and colorectal cancers is shown to impact negatively on survival [49], while in oesophageal adenocarcinoma, ICOS alone, and in the presence of CD45RO, was found to significantly improve 5 year OS [19].

Based on our findings and those of others highlighted, we observe that there is an agreement with regards to expressions of TILs across gastrointestinal (GI) tumours and their

prognostic significance. However, there are also conflicting reports with regards to the presence of ICIs and their impact on survival outcome in the various tumour types. We have previously reported the molecular changes and genetic mutations associated with SBA with their therapeutic implications and clinical potential as prognostic biomarkers [2]. For immune checkpoint therapy to be successfully implemented especially in SBAs, it will be beneficial to consider multiple ICI strategies targeting more than one immune checkpoint protein. As we have shown, high levels of PD-L1 were associated with high levels of other checkpoint proteins (IDO1 and ICOS). Therefore, use of anti-PD-1 therapy in combination with other ICI therapeutics may yield significantly more effective treatment response in a subset of SBA patients.

In the study of GI tract tumours, there are inconsistencies in the literature regarding the characterisation of the small intestine microenvironment, and the expression of molecular markers and their prognostic significance. These discrepancies may be due to factors such as sample variability, tumour heterogeneity, differences in experimental methodologies, sample size and statistical power, patient characteristics, and confounding factors. The tumour microenvironment and tumour heterogeneity can contribute to variations in molecular marker expression patterns. Differences in sample collection, processing, and analysis, as well as patient characteristics, can also influence results. Confounding factors, such as treatment history, disease stage, and genetic variability, may also contribute to variability. Future studies should aim to account for these factors to improve the accuracy and reproducibility of molecular marker analyses in GI tract tumours. We advocate for a national consortium effort for the standardised and consistent collection of rare tumours, including SBA.

We acknowledge that the limited number of cases within our study suggests a degree of cautiousness should be applied with the interpretation of our findings. However, the representativeness of our data is associated with the fact that, SBA is a rare cancer, coupled with the especially low number of cases in Northern Ireland. This observation is particularly relevant when discussing the prognostic impact on survival of PD-L1 positivity in SBA. There is ample evidence demonstrating that PD-L1 positivity negatively impacts survival in SBA [50, 51], this is far from conclusive, as has been commented on by others [14]. For example, our data correlates with larger studies such as Giuffrida et al. (n = 121 cases) and of Klose et al. (n = 75 cases) both of whom also report PD-L1 positivity being associated with prolonged survival [13, 52]. Additionally, Wirta et al. (n = 96 cases) also showed High PD-L1 was prognostic for better patient outcome. Our data while limited, add to the paucity of good research data on SBA, particularly in terms of recognising risk factors, establishing pathogenesis as well as creating prevention and awareness programmes at a global level [53, 54].

## Conclusion

With this study, we describe the immune and immune checkpoint landscape in a small cohort of small bowel adenocarcinoma patients. We are able to show the presence of lymphocytes and immune checkpoint markers especially PD-L1, ICOS and IDO1 within tumour samples and their significance in terms of patient survival. We also identify a correlation between expression of PD-L1 and IDO1 and based on this, propose a potential approach to the implementation of immune checkpoint therapy as well as patient stratification in the management of SBA.

## Supporting information

**S1 File.**
(XLSX)

## Acknowledgments

The authors wish to acknowledge the Northern Ireland Biobank for tissues used in this study under NIB13-0067.

## Author Contributions

**Conceptualization:** Stephen McQuaid, Matthew P. Humphries, Manuel Salto-Tellez.

**Data curation:** Fatima Abdullahi Sidi, Victoria Bingham, Stephanie G. Craig, Matthew P. Humphries.

**Formal analysis:** Victoria Bingham, Stephanie G. Craig, Matthew P. Humphries.

**Investigation:** Matthew P. Humphries.

**Methodology:** Fatima Abdullahi Sidi, Victoria Bingham, Stephanie G. Craig, Matthew P. Humphries.

**Project administration:** Jacqueline A. James, Matthew P. Humphries, Manuel Salto-Tellez.

**Resources:** Richard C. Turkington, Jacqueline A. James, Manuel Salto-Tellez.

**Supervision:** Stephen McQuaid, Richard C. Turkington, Jacqueline A. James, Matthew P. Humphries, Manuel Salto-Tellez.

**Validation:** Victoria Bingham, Stephen McQuaid, Matthew P. Humphries, Manuel Salto-Tellez.

**Visualization:** Stephen McQuaid, Matthew P. Humphries.

**Writing – original draft:** Stephen McQuaid, Matthew P. Humphries, Manuel Salto-Tellez.

**Writing – review & editing:** Stephen McQuaid, Richard C. Turkington, Jacqueline A. James, Matthew P. Humphries, Manuel Salto-Tellez.

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
