## [Decision Letter · Decision Letter 0]

7 Mar 2023

PONE-D-22-31468Exploring the immune microenvironment in small bowel adenocarcinoma using digital image analysis.PLOS ONE

Dear Dr. Humphries,

Thank you for submitting your manuscript to PLOS ONE. After careful consideration, we feel that it has merit but does not fully meet PLOS ONE’s publication criteria as it currently stands. Therefore, we invite you to submit a revised version of the manuscript that addresses the points raised during the review process.

We look forward to receiving your revised manuscript.

Kind regards,

Vincenzo L'Imperio, MD

Academic Editor

PLOS ONE

Journal Requirements:

2. We note that the grant information you provided in the ‘Funding Information’ and ‘Financial Disclosure’ sections do not match.When you resubmit, please ensure that you provide the correct grant numbers for the awards you received for your study in the ‘Funding Information’ section.

“This study was funded by a Cancer Research UK (CRUK) Accelerator Grant (A20256) to JJ and MST. CRUK (https://www.cancerresearchuk.org/) had no role in study design, data collection and analysis, decision to publish, or preparation of the manuscript. The Northern Ireland Biobank has received funds from HSC Research and Development Division of the Public Health Agency in Northern Ireland”

“This study was funded by a Cancer Research UK (CRUK) Accelerator Grant (A20256) to JJ and MST. CRUK (https://www.cancerresearchuk.org/) had no role in study design, data collection and analysis, decision to publish, or preparation of the manuscript. The Northern Ireland Biobank has received funds from HSC Research and Development Division of the Public Health Agency in Northern Ireland.”

“I have read the journal's policy and the authors of this manuscript have the following competing interests: Manuel Salto-Tellez is a scientific advisor to Mindpeak and Sonrai Analytics, and has received honoraria recently from BMS, MSD and Incyte. None of these disclosures are related to this work. The remaining authors declare no potential conflicts of interest.”

6.We note that you have indicated that data from this study are available upon request. PLOS only allows data to be available upon request if there are legal or ethical restrictions on sharing data publicly. For more information on unacceptable data access restrictions, please see http://journals.plos.org/plosone/s/data-availability#loc-unacceptable-data-access-restrictions.

Reviewers' comments:

Reviewer's Responses to Questions

**Comments to the Author**

1. Is the manuscript technically sound, and do the data support the conclusions?

Reviewer #1: Yes

Reviewer #2: Yes

2. Has the statistical analysis been performed appropriately and rigorously? 

Reviewer #1: Yes

Reviewer #2: Yes

3. Have the authors made all data underlying the findings in their manuscript fully available?

Reviewer #1: Yes

Reviewer #2: Yes

4. Is the manuscript presented in an intelligible fashion and written in standard English?

Reviewer #1: Yes

Reviewer #2: Yes

5. Review Comments to the Author

Reviewer #1: The paper describes the immunohistochemical features of the inflammatory infiltrate in 25 cases of small bowel adenocarcinomas and their relationship to long-term endpoints such as overall survival and progression-free survival. Similar studies have been conducted throughout the gastrointestinal tract, including the small bowel, but the low incidence of the disease and the wide panel of markers tested make this study interesting.

The structure of the paper could be improved by making the introduction less generic and repetitive and presenting the topics covered more specifically. Additionally, some methodological choices should be better clarified and justified, both in the general and computational parts. Furthermore, some additional information could add value to the study, starting from the data already collected.

More specifically, in the introduction (line 53), the sentence beginning with “although rare” is a bit confusing, it is not clear whether the authors want to underline the fact that it is a very rare neoplasm or in some way not so irrelevant.

In line 56, the authors state that incidence rates have increased by 151%, but a brief explanation of this relevant epidemiological change is missing. In the same line, the authors state that SBC occurs more frequently in females than males, but the reference provided shows a slight male predominance.

In line 61, the authors state that the management of SBA is similar to that of CRC due to similar pathogenesis. However, it is unclear how pathogenesis impacts treatment, which is mainly surgical for both malignancies. Moreover, in the paragraph starting at line 81, treatment options seem to be slightly different between these malignancies. In line 63, differences in incidence and disease progression between SBA and CRC are mentioned, but the causes are not specified. Overall, the general comparison between SBA and CRC seems to be somewhat out of context. More focus should be given to microenvironment in SBA and if desired in the other districts of the gastrointestinal tract.

In the paragraph starting at line 81, the authors provide some information on treatment, prognosis, and molecular characteristics. However this part is too generic and is not preparatory to the study carried out in this paper. I would focus on which methods are most successful in studying the microenvironment, how immunohistochemistry has been used in this setting and how an image computational approach impacts immunohistochemistry evaluation and quantification in pathology and research.

In Material and Methods, line 106, the authors describe how TMA was constructed, but the concept of "representative area" is not clarified. This could represent a major criticism of the work because the IHC quantitative analysis could undergo significant changes due to tumor heterogeneity. Therefore, an exhaustive explanation is needed.

In line 121, the authors should explain how artifacts have been removed (manually?), as the reference don’t explain this.

In line 127, the authors should better explain how the tumor-stroma classifier has been applied and which QuPath variables and parameters have been used. It is also necessary to specify who performed the annotations. Treshold for positive/negative immunohistochemistry has been adjusted for each image or a standard treshold has been used? How did authors tackled the problem of stain variability?

The automatic identification of IHC positivity may be quite intuitive when the evaluation is based on nuclear positivity. However, some of these markers have membrane staining, which is much more challenging. The workflow followed to deal with this task should be provided.

In the section on Data and Statistical Analysis, the authors should provide an explanation for why they used “average scores” instead e.g. of “maximum scores” for their analysis, and why they dichotomized density rather than treating it as a continuous variable. It would also be helpful to know if previous studies have employed these same methods.

In the Results section, it would be beneficial if the authors integrated clinicopathological data with the pathologist's evaluation of infiltrate, as is usually done in the pathological report of colorectal cancer, in order to provide a quantitative evaluation of intratumoral and peripheral infiltrate.

In Biomarker Scoring, the authors only define results in terms of absolute density. It may also be informative to include relative quantification, such as comparing the proportion of CD4/CD8 T cells or lymphocytic versus histiocytic infiltrate. For example, determining if a tumor with a CD4/CD8 ratio higher than a certain cutoff has better or worse endpoints than one with a different value could add significant value to this study.

The Discussion section offers interesting ideas and interesting comparisons with the literature. A further strong point of the study would be to clarify or at least hypothesize how there are discrepancies in the literature on the study of the microenvironment at the small intestine level, how there are differences along the gastrointestinal tract (briefly) and how some markers have significant results only at one of the two endpoints.

Reviewer #2: The introduction is a bit long. Some of these info could be moved to the discussion.

L86: sentence starts with "While" but there is just a single clause. Perhaps you meant ", while [...]" ?

Just out of curiosity, is there a specific reason why an older (v0.2.2) version of QuPath was used?

L123: you're missing a hyphen in "pathologist-guided"

The QuPath part in the methods is described a bit opaquely and it contrasts with the rest of the section (e.g. the IHC is described in great detail). I suggest adding more textual details and/or pictures to make the process more understandable and hopefully reproducible.

On the same topic, L130 hides in a line what might have deserved its own paragraph. How did this manual check (pathologist confirmation) take place? Was it done extensively for all markers for all cases, or just on a sample? Was the classification in QuPath good enough to base all further analyses on it?

Table 2. Tumour site in table does not match the preceding text (DJ junction 1 vs 2; NOS 4 vs 3).

L171. 8423 CD3+ cells in a mm2 is a lot. It roughly amounts to that mm2 being essentially filled by small lymphocytes. Was this a very briskly inflamed tumor? Were we in a lymph node? Was any viable tumor present? Was the 8423 cells/mm2 calculation correct?

L213: small typo: you either assessed the expression of immune cell and macrophage _markers_ (CD3, etc), or you assessed the _presence_/quantity of immune cells and macrophages

L216 are you using "stromal" as a synonym for "inflammatory cell"/"lymphocyte"? I think the latter would be clearer.

Figure 3. What data, exactly, were tested for correlation? The absolute number of cells positive for each marker (except for PDL1)? This would explain the homogenously very high correlation coefficients observed and would suggest that this figure does not add much to the manuscript. For instance, CD4 and CD8 are expressed in addition to CD3 in T lymphocytes, and so it is only expected that this strong correlation is observed.

6. PLOS authors have the option to publish the peer review history of their article (what does this mean?). If published, this will include your full peer review and any attached files.

Reviewer #1: **Yes: **Giorgio Cazzaniga

Reviewer #2: **Yes: **Alessandro Caputo

---

## [Author Response · Author response to Decision Letter 0]

3 May 2023

Reviewers' comments:

We thank the reviewers for their kind words and keen observations. We believe we have addressed the points raise.

Reviewer #1: 

Reviewer :

The paper describes the immunohistochemical features of the inflammatory infiltrate in 25 cases of small bowel adenocarcinomas and their relationship to long-term endpoints such as overall survival and progression-free survival. Similar studies have been conducted throughout the gastrointestinal tract, including the small bowel, but the low incidence of the disease and the wide panel of markers tested make this study interesting.

The structure of the paper could be improved by making the introduction less generic and repetitive and presenting the topics covered more specifically. Additionally, some methodological choices should be better clarified and justified, both in the general and computational parts. Furthermore, some additional information could add value to the study, starting from the data already collected.

Author response: We thank the reviewer for this direction regarding specific presentation. We have broken the introduction in to clearer sub-headings in the introduction. Aetiology and incidence; The tumour microenvironment in gastrointestinal tumour biology; Pages 3-4.

We have additionally condensed the text to reduce any repetition. Page 3. Line 57-62. Following a comment below to focus less on treatment, prognosis, and molecular characteristics, we have removed text. Page 4. Line 91-97. Additional information has been added throughout, as indicated by the specific reviewer points addressed below which add further value as suggested. 

Reviewer:

More specifically, in the introduction (line 53), the sentence beginning with “although rare” is a bit confusing, it is not clear whether the authors want to underline the fact that it is a very rare neoplasm or in some way not so irrelevant.

Author response: Thank you for highlighting this possible misinterpretation. We have edited the text for clarity. Page 3. Line 58

Reviewer:

In line 56, the authors state that incidence rates have increased by 151%, but a brief explanation of this relevant epidemiological change is missing. In the same line, the authors state that SBC occurs more frequently in females than males, but the reference provided shows a slight male predominance.

Author response: We value the reviewer’s keen observations. Since our original draft, national incidence statistics have been updated. We have altered the text to reflect this, inclusive of the removal of reference 3, and the inclusion of 4 additional references for the incidence rates in England, Wales, Scotland and Northern Ireland (References 3-6). Page 3. Line 57-62.

The reasons proposed for this epidemiological change have now been included. Additional text has been included. Page 3. Line 62-64. Additional references are included (References 7-11). 

Reviewer:

In line 61, the authors state that the management of SBA is similar to that of CRC due to similar pathogenesis. However, it is unclear how pathogenesis impacts treatment, which is mainly surgical for both malignancies. Moreover, in the paragraph starting at line 81, treatment options seem to be slightly different between these malignancies. In line 63, differences in incidence and disease progression between SBA and CRC are mentioned, but the causes are not specified. Overall, the general comparison between SBA and CRC seems to be somewhat out of context. More focus should be given to microenvironment in SBA and if desired in the other districts of the gastrointestinal tract.

Author response: We would agree that pathogenesis and cancer management do not necessarily imply similar treatment courses. We have modified the sentence to prevent any assumed association between treatment and pathogenesis. Page 4. Line 69-70.

The proposed causes affecting the incidence and progression of the disease have now been specified with additional references. Page 3. Line 72-73. References 15 & 17.

Reviewer:

In the paragraph starting at line 81, the authors provide some information on treatment, prognosis, and molecular characteristics. However this part is too generic and is not preparatory to the study carried out in this paper. I would focus on which methods are most successful in studying the microenvironment, how immunohistochemistry has been used in this setting and how an image computational approach impacts immunohistochemistry evaluation and quantification in pathology and research.

Author response: We accept the comment treatment, prognosis, and molecular characteristic text may not give the paper a clear narrative. We have therefore added text to comment on the methods which are considered the most successful in studying the TME of SBA. We include flow cytometry, and gene expression profiling. We also underscore the key IHC papers of Wirta et al. and Liu et al. Page 4. Line 91-97. (References 14, 15, 23, 24). 

Though we feel a discussion on ‘how an image computational approach impacts immunohistochemistry evaluation and quantification in pathology and research’ is an important topic, we feel this is a very board subject worthy of its own separate publication. We trust the reviewer would consider our wish not to explore this question here, and consider this out of scope. We hope our revisions have kept the content as focused on SBA data as possible, as requested. 

Reviewer:

In Material and Methods, line 106, the authors describe how TMA was constructed, but the concept of "representative area" is not clarified. This could represent a major criticism of the work because the IHC quantitative analysis could undergo significant changes due to tumor heterogeneity. Therefore, an exhaustive explanation is needed.

Author response: This is a key consideration in the creation of a TMA and we thank the reviewer for labouring this point, which is often over looked. We go to significant lengths to ensure our TMAs are robust and representative. To this end we have included an exhaustive comment on the steps and thought process of our ‘representative tumour selection’. Page 5-6. Line 124-130.

Reviewer:

In line 121, the authors should explain how artifacts have been removed (manually?), as the reference don’t explain this.

Author response: We are happy to clarify this was indeed conducted manually. Page 8. Line 145. 

Reviewer: 

In line 127, the authors should better explain how the tumor-stroma classifier has been applied and which QuPath variables and parameters have been used. It is also necessary to specify who performed the annotations. Treshold for positive/negative immunohistochemistry has been adjusted for each image or a standard treshold has been used? How did authors tackled the problem of stain variability?

Author response: The reviewer rightly requests more detail on the specifics of the tumour-stroma classifier variables and parameters used. We have included a detailed summary of the features used to build the tumour-stroma classifier as well as to indicate the individuals performing the annotations and reviewing the models accuracy. Page 8. Line 153-160.

The thresholding level was standardised across all images. This is an important point to include as a measure of consistency in the analysis portion of the study. Many publications do not adhere to this and we thank the reviewer for requesting we confirm this. We have indicated this in the text. Page 8. Line 149-151

Staining variability is also a factor we rigorously control within this, and all our studies. Again the reviewer is right to request clarity on this. Our staining is conducted on automated staining platforms ensuring consistency, efficiency, standardization, quality control, and minimal operator variability. Where possible staining is conducted in a single batch to reduced variability further. We have added a comment on our use of an automated stainer in this study, to ensure staining consistency. Page 6. Line 134-132. 

Reviewer:

The automatic identification of IHC positivity may be quite intuitive when the evaluation is based on nuclear positivity. However, some of these markers have membrane staining, which is much more challenging. The workflow followed to deal with this task should be provided.

Author response: This comment is from a very experienced perspective, and we wholeheartedly agree with the challenging aspect of membrane analysis by digital pathology, particularly QuPath. We have expounded this process and provided a detailed analysis workflow in previous work and are happy to reference it here for further reading. Page 8. Line 163-164. (References 29-31). 

Reviewer: 

In the section on Data and Statistical Analysis, the authors should provide an explanation for why they used “average scores” instead e.g. of “maximum scores” for their analysis, and why they dichotomized density rather than treating it as a continuous variable. It would also be helpful to know if previous studies have employed these same methods.

Author response: We are glad to provide detail on our approach here. As our TMAs are representative of the tumour area, and are created in triplicate, our decision to average our data ensures the values reported are a true reflection of the patient tumour. As far as is possible, averaging our data controls for any influence by spatial or topographical heterogeneity which may provide a skewed result if using a maximum score, for example. This approach is very much in the spirit of the reviewer’s earlier question, regarding representative tumour areas and ensuring consistency in the analysis. 

With regards to the dichotomisation of our density data. Cut off at the median is considered acceptable practice when reporting IHC data. The use of a median values again points to our goal of producing representative data. We are experienced with alternative dichotomisation methods such as, ROC curves or distribution based cut offs. However, these often require very large data sets to be considered robust. We acknowledge the paucity of our dataset and as such the use of, for example, the Euclidian distance measured cut-off by ROC curve, would cherry pick the ‘best’ p-value from our limited ‘less smooth’ data. We are confident that any significance values obtained by median cut-off would be even more significant by other methods and we do not wish to over interpret our data. Additionally, categorical data was required for the generation of Kaplan-Meier curves. 

Continuous variable analysis would have been necessary had we conducted a multivariate analysis, inclusive of all clinical data. This was a decision we did not take, again due to the limited data with the study, which may have led to an over interpretation of the data. 

Reviewer:

In the Results section, it would be beneficial if the authors integrated clinicopathological data with the pathologist's evaluation of infiltrate, as is usually done in the pathological report of colorectal cancer, in order to provide a quantitative evaluation of intratumoral and peripheral infiltrate.

Author response: We accept there is lacking context regarding the pathological description of the whole specimen from which the TMA was made. The goal of our study was to add to the collective knowledge of molecular data from tumour core samples in SBA. While we appreciate the influence of tumour infiltrate in regards to prognostic outcome in patients with GI cancers, this data is beyond the scope of our study. We fully intend to conduct a larger descriptive study on whole slide images of SBAs, which will necessitate the descriptive data analysis the reviewer is highlighting. 

Reviewer: 

In Biomarker Scoring, the authors only define results in terms of absolute density. It may also be informative to include relative quantification, such as comparing the proportion of CD4/CD8 T cells or lymphocytic versus histiocytic infiltrate. For example, determining if a tumor with a CD4/CD8 ratio higher than a certain cutoff has better or worse endpoints than one with a different value could add significant value to this study.

Author response: The suggest analysis was undertaken by the team to probe the dataset further. However, when dichotomised in to corresponding high and low categories for CD4/CD8, our data was reduced to 15 cases (8 high, 7 low). Analysis of these data did not reveal any significance of note, and was not included in the manuscript. Our averaged data has been made available for further analysis. Page 9. Line 174-175.

Reviewer:

The Discussion section offers interesting ideas and interesting comparisons with the literature. A further strong point of the study would be to clarify or at least hypothesize how there are discrepancies in the literature on the study of the microenvironment at the small intestine level, how there are differences along the gastrointestinal tract (briefly) and how some markers have significant results only at one of the two endpoints.

Author response: We agree the discussion of these discrepancies could add to the conversation surrounding the significance of molecular data from various GI tumour tissue origins. To this end we have included an additional paragraph discussing the factors which may influence the wide ranging, and sometimes contradictory, data reported in the literature. Page 17. Line 337-348.

Reviewer #2: 

Reviewer: 

The introduction is a bit long. Some of these info could be moved to the discussion.

Author response: We thank the reviewer for this direction regarding specific presentation, which follows a similar comment from reviewer 1. We have now broken the introduction in to clearer sub-headings in the introduction. Aetiology and incidence; The tumour microenvironment in gastrointestinal tumour biology; Pages 3-4

We have additionally condensed the text to reduce any repetition. Page 3. Line 57-62. Following a comment below to focus less on treatment, prognosis, and molecular characteristics, we have removed text. Page 4. Line 91-97. Additional information has been added to the discussion as advised by reviewer 1. We believe this has enhanced our manuscript. Page 17. Line 337-348.

Reviewer: 

L86: sentence starts with "While" but there is just a single clause. Perhaps you meant ", while [...]" ?

Author response: This sentence has now been removed from the manuscript following reviewer feedback. 

Reviewer: 

Just out of curiosity, is there a specific reason why an older (v0.2.2) version of QuPath was used?

Author response: The main driver of the use of QuPath v0.2.2 within this paper is mainly due to the date at which our image analysis of these data began. While we have since upgraded our tools to newer versions of QuPath it was beneficial for consistency to keep our analysis for this project confined to one version of software. Furthermore, the older version of the software aided in our compatibility of established scripts which sped up our analysis, we also had legacy institution system requirements, as well as specific user interface customisations.

Reviewer: 

L123: you're missing a hyphen in "pathologist-guided"

Author response: Thank you for this note. This has been rectified. Page 8. Line 147.

Reviewer: 

The QuPath part in the methods is described a bit opaquely and it contrasts with the rest of the section (e.g. the IHC is described in great detail). I suggest adding more textual details and/or pictures to make the process more understandable and hopefully reproducible.

Author response: We realise this has been an overlooked factor in our methodology reporting. As highlighted by another reviewer to this end we have expanded our explanation of the digital pathology workflow and process we have followed. The reviewer rightly requests more detail on the specifics of the tumour-stroma classifier variables and parameters used. We have included a detailed summary of the features used to build the tumour-stroma classifier as well as indicating the individuals performing the annotations and reviewing the models accuracy. Page 8. Line 153-160

The thresholding level was standardised across all images. This is an important point to include as a measure of consistency in the analysis portion of the study. Many publications do not adhere to this and we thank the reviewer for requesting we confirm this. We have indicated this in the text. Page 8. Line 149-151.

Staining variability is also a factor we rigorously control within this, and all our studies. Again the reviewer is right to request clarity on this. Our staining is conducted on automated staining platforms ensuring consistency, efficiency, standardization, quality control, and minimal operator variability. Where possible staining is conducted in a single batch to reduced variability further. We have added a comment on our use of an automated stainer in this study, to ensure staining consistency. Page 6. Line 134-135. 

Reviewer: 

On the same topic, L130 hides in a line what might have deserved its own paragraph. How did this manual check (pathologist confirmation) take place? Was it done extensively for all markers for all cases, or just on a sample? Was the classification in QuPath good enough to base all further analyses on it?

Author response: We are grateful to all reviewers for requesting more detail regarding pathologist review, IHC thresholding, and QuPath classification. Pathologist review was conducted on all markers across multiple images. We have indicated this in the text. Page 8. Line 149-151. We also point the reader to our previous publications where we describe in further detail our process of pathologist review. Page 8. Line 163-164. 

In line with the point above, we have included a detailed summary of the features used to build the tumour-stroma classifier as well as indicate the individuals performing the annotations and reviewing the models accuracy. The classification was consistently applied across all images. Page 8. Line 153-160.

Reviewer: 

Table 2. Tumour site in table does not match the preceding text (DJ junction 1 vs 2; NOS 4 vs 3).

Author response: Apologies for the oversight in the text. We have confirmed the clinical details and amended the text to reflect the case details. Page 9. Line 185.

Reviewer: 

L171. 8423 CD3+ cells in a mm2 is a lot. It roughly amounts to that mm2 being essentially filled by small lymphocytes. Was this a very briskly inflamed tumor? Were we in a lymph node? Was any viable tumor present? Was the 8423 cells/mm2 calculation correct?

Author response: We agree the number of CD3 cells is high, and this did initially gave us pause, however we can assure the reviewer that this TMA core does indeed contain a highly inflamed tumour sample. As part of our due diligence we reviewed the H&E and confirmed the tumour presence in the highly inflamed specimen which correlated with the high CD3 count. 

Reviewer: 

L213: small typo: you either assessed the expression of immune cell and macrophage _markers_ (CD3, etc), or you assessed the _presence_/quantity of immune cells and macrophages

Author response: Thank you pointing this out. We have been more specific in the text. Page 14. Line 272-273. 

Reviewer: 

L216 are you using "stromal" as a synonym for "inflammatory cell"/"lymphocyte"? I think the latter would be clearer.

Author response: We have modified the text for clarity. Page 14. Line 275. 

Reviewer: 

Figure 3. What data, exactly, were tested for correlation? The absolute number of cells positive for each marker (except for PDL1)? This would explain the homogenously very high correlation coefficients observed and would suggest that this figure does not add much to the manuscript. For instance, CD4 and CD8 are expressed in addition to CD3 in T lymphocytes, and so it is only expected that this strong correlation is observed.

Author response: The absolute number for positive cells for each marker, except PD-L1 was used, as the reviewer asks. We have added this to the figure 3 legend for further clarity. Page 13. Line 255-257.

We agree the correlation of T lymphocytes is an obvious and expected correlation. However, we feel the correlation of markers figure is important in the manuscript as it informs and directs much of our discussion. We did note some interesting and unexpected correlations. For instance between PD-L1 and 1DO-1, which we discuss at length. Also, we observe strong correlations of T cells with PD-L1, this is in similarity to other studies which have made such observations. Our data here adds to the collective domain knowledge of these associations.

---

## [Editor Report · Decision Letter 1]

18 Jul 2023

Exploring the immune microenvironment in small bowel adenocarcinoma using digital image analysis.

PONE-D-22-31468R1

Dear Dr. Humphries,

We’re pleased to inform you that your manuscript has been judged scientifically suitable for publication and will be formally accepted for publication once it meets all outstanding technical requirements.

Kind regards,

Vincenzo L'Imperio, MD

Academic Editor

PLOS ONE

Additional Editor Comments:

The authors adequately addressed all the reviewers' comments improving the quality and value of their manuscript in its present form.

---

## [Editor Report · Acceptance letter]

21 Jul 2023

PONE-D-22-31468R1 

Exploring the immune microenvironment in small bowel adenocarcinoma using digital image analysis. 

Dear Dr. Humphries:

I'm pleased to inform you that your manuscript has been deemed suitable for publication in PLOS ONE. Congratulations! Your manuscript is now with our production department. 

Kind regards, 

on behalf of

Dr. Vincenzo L'Imperio 

Academic Editor

PLOS ONE